# Barriers to the Provision of Preventive Care to People Living with Mental Health Conditions: Self-Report by Staff Working in an Australian Community Managed Organisation

**DOI:** 10.3390/ijerph19084458

**Published:** 2022-04-07

**Authors:** Tara Clinton-McHarg, Lauren Gibson, Kate Bartlem, Sonya Murray, Jade Ryall, Mark Orr, Janet Ford, Jenny Bowman

**Affiliations:** 1School of Psychology, The University of Newcastle, Callaghan, NSW 2308, Australia; tara.clinton-mcharg@newcastle.edu.au (T.C.-M.); lauren.k.gibson@uon.edu.au (L.G.); kate.bartlem@newcastle.edu.au (K.B.); 2The Australian Prevention Partnership Centre, Sax Institute, Ultimo, NSW 2037, Australia; 3Hunter Medical Research Institute, New Lambton Heights, NSW 2305, Australia; sonya.murray@hmri.org.au; 4Priority Research Centre for Health Behavior, The University of Newcastle, Callaghan, NSW 2308, Australia; 5Flourish Australia, Sydney Olympic Park, NSW 2127, Australia; jade.ryall@flourishaustralia.org.au (J.R.); mark.orr@flourishaustralia.org.au (M.O.); janet.ford@flourishaustralia.org.au (J.F.)

**Keywords:** mental health, preventive care, barriers, community organization, chronic disease

## Abstract

People living with mental health conditions experience a reduced life expectancy largely due to a higher prevalence of chronic diseases. Addressing health risk behaviours, including tobacco smoking, inadequate nutrition, harmful alcohol consumption, and physical inactivity (SNAP), through the provision of preventive care, is recommended to reduce this burden. Community Managed Organisations (CMOs) may play an important role in providing preventive care to consumers with mental health conditions, however, few studies have examined preventive care provision in CMO settings; and no studies have comprehensively assessed barriers to the provision of this care using a tool such as the Theoretical Domains Framework (TDF). To fill this research gap, we conducted an online survey among staff (N = 190) from one CMO in Australia to (1) identify barriers to preventive care provision (ask, advise, assist, connect) to address SNAP behaviours among consumers; and (2) explore associations between barriers and preventive care provision. Results demonstrate that while staff reported knowing how to provide preventive care and believed it would positively impact consumers; barriers including confidence in providing this care and consumer uptake of referrals, were identified. Further research among multiple CMOs is needed to identify care provision and associated barriers in the sector more widely.

## 1. Introduction

Globally, people living with mental health conditions have a higher prevalence of physical chronic disease (e.g., cardiovascular disease, cancer, diabetes) compared to the general population [1,2,3,4]. In Australia, this higher prevalence of chronic disease accounts for around 70% of the difference in life expectancy for people with mental health conditions, estimated to be 16 years less for males, and 12 years less for females [4]. Modifiable health risk behaviours including tobacco smoking, inadequate nutrition, harmful alcohol consumption, and physical inactivity are reported to be more prevalent among people with mental health conditions compared to those without [5,6,7,8]. Many people with mental health conditions desire to change their health behaviours [9,10,11,12] and can successfully do so if adequate support is provided [13]. Therefore, routine delivery of preventive care has been recommended to help reduce the prevalence of chronic disease risk factors among people with mental health conditions [14,15,16,17].

Brief interventions are one form of preventive care that has been shown to be practical in situations where staff are unable to provide more intensive support [18]. Evidence-based models of a brief intervention to address chronic disease risk behaviours include the 5As model (Ask, Assess, Advise, Assist, Arrange) [19,20,21,22] and the briefer AAC model (Ask, Advise, and Connect) [23]. Existing clinical guidelines also acknowledge that preventive care is a vital component of holistic care for people with mental health conditions [14,15,16,17]. Research has found that receiving assessments, brief advice, and follow-up referrals can assist individuals with mental health conditions to modify their health risk behaviours [13,24,25]. However, despite these guidelines, the provision of preventive care by health service staff to people with mental health conditions is often reported to be suboptimal [24], with competing clinical priorities, large clinical loads, and time constraints [25,26,27,28] reported as barriers to preventive care delivery.

In addition to clinical mental health services, there are other settings where preventive care may have the potential to be delivered to people with mental health conditions. Community Managed Organisations (CMOs) are not-for-profit organisations that have been established to offer psychosocial support services to people with mental health conditions and their carers [29]. CMOs usually work alongside the clinical healthcare system to address a broad range of needs for people with mental health conditions, for example providing transport, accommodation, peer support, and education [29]. The role that CMOs are playing in the mental health sector is increasing, with funding for mental health psychosocial support services growing 10-fold over the past 25 years [30]. People with mental health conditions accessing CMO services (i.e., consumers) often rate their experiences with CMOs positively. One survey of consumers from 10 different CMOs in New South Wales, Australia found that 89% of participants (N = 1058) rated their experience with their CMO as “excellent” or “very good” [31]. Given the increasing reach of CMOs to people with mental health conditions, and their focus on improving psychosocial wellbeing more broadly, CMOs may be an opportune setting in which the chronic disease risk behaviours of consumers could be addressed [17,32,33,34].

There is some evidence that at least part of the 5As and AAC models form existing policies and guidelines for CMOs [33,34,35]. For example, one study of CMO leaders in New South Wales [33] found that smoking cessation support was provided to consumers through: *assessment* of smoking status, information about smoking cessation supports available (i.e., *advise*), and referral to external support for smoking cessation (i.e., *connect*). Additional support to *assist* consumers with smoking cessation was also reported, including support groups, Nicotine Replacement Therapy, and “SANE Smoke Free Kits”. However, across CMOs there is no uniform policy or guideline for addressing health risk behaviours and each individual organisation may design its own policies or guidelines.

Although the opportunity exists for CMOs to address health risk behaviours among people with mental health conditions, only a small number of Australian studies have examined the level of preventive care currently being provided in this setting [33,34,35,36]. This previous research suggests that low levels of preventive care support are being provided to consumers accessing CMOs [34]. In order to understand how to best support CMOs to provide optimal preventive care to their consumers, it is important to identify what factors might be acting as barriers to the provision of such care. One study conducted with 35 CMOs in New South Wales Australia in 2014 reported that a lack of funding, a lack of resources, and staff workloads were the primary barriers to CMOs addressing the physical health of consumers [34]. However, a comprehensive assessment of barriers (using a specific implementation framework) was not conducted and the term “physical health” was not specifically defined. Another study, conducted with 38 CMOs in New South Wales (2009), found that the main barriers to staff providing smoking cessation care were: lack of time; feeling that smoking cessation support was not a priority; not knowing where to refer consumers for support; and having a lack of policies in the service that provided clear direction on how to address smoking. However, once again this study did not comprehensively assess possible barriers, and only interviewed a small sample of staff [36].

Internationally, few studies have explored barriers to CMOs addressing health risk behaviours among people with mental health conditions [37,38]. One qualitative study of leaders of US psychiatric rehabilitation services ([38] comparable to CMOs in Australia) identified barriers to providing physical health support to include difficulties coordinating with external providers, using health IT, and finding staff who were competent and enthusiastic about providing physical health support to consumers. However, this study did not comprehensively assess possible barriers and interviewed leaders of organisations rather than staff providing direct support to consumers. Only one study internationally has assessed CMO staff perceptions of facilitators or barriers to preventive care provision for a health risk behaviour [37]. This study focused on tobacco smoking and was conducted among 605 staff working across multiple CMOs in New Zealand. Factors identified as facilitating CMO staff to address smoking among consumers included: receiving training, having organisational policies, and having a staff culture that promoted positive lifestyle behaviours. Across previous research conducted within Australia and internationally, no study has comprehensively assessed CMO staff reported barriers to the provision of preventive care for multiple health risk behaviours for people living with mental health conditions.

One tool that can be used to comprehensively identify potential barriers and facilitators to the implementation of recommended practices is the Theoretical Domains Framework (TDF). The TDF was developed to consolidate constructs from various theories on behaviour change and implementation into a comprehensive set of domains [39]. Whilst the TDF has previously been used to identify barriers to the provision of preventive care in clinical health care settings [40,41] no studies to date have used the tool to identify such barriers and facilitators in CMO settings. The TDF is useful for identifying areas of need for staff training, resource development, organisational culture, and guideline and policy development. This information can also inform the development of implementation interventions that may help to overcome reported barriers and increase preventive care provision in CMOs.

Therefore, the aims of the current study were: (1) to use the TDF to identify potential barriers to providing elements of preventive care (ask, advise, assist, connect) to address health risk behaviours (smoking, low fruit and vegetable consumption, harmful alcohol consumption, physical inactivity) in people with mental health conditions; and (2) to explore associations between potential barriers to preventive care provision, and CMO staff members’ reported provision of care (i.e., providing connections to support services) for consumers with health risk behaviours (smoking, low fruit and vegetable consumption, harmful alcohol consumption, physical inactivity).

## 2. Materials and Methods

### 2.1. Design and Setting

A cross-sectional, online survey of all staff members was conducted within one large CMO, which has approximately 60 service sites in both New South Wales and Queensland, Australia [42]. The CMO delivers a range of support services to people living with mental health conditions such as housing initiatives, connection to relevant services external to the CMO, vocational support, social support, community outreach support, and assistance for family members and carers. At the time of the survey, the CMO had been established for more than 60 years, employed 673 staff, and provided care to almost 5000 consumers per annum [42]. The study was approved by the University of Newcastle Human Research Ethics Committee (H-2016-0401).

### 2.2. Participants

CMO staff currently employed by the CMO were eligible to complete the survey if they: (1) had been employed by the CMO for more than two months; and (2) either provided direct care to CMO consumers (people living with a mental health issues) who were 18 years of age or older or supervised or managed staff who provided direct care to such consumers.

### 2.3. Recruitment and Data Collection

Potential participants were identified by the CMO and invited to take part in the research via an email sent by a senior CMO staff member, which included an information statement and a link to an online survey. Following the original email, reminder emails were sent one and three weeks later. The online survey was programmed and administered using the survey platform Qualtrics [43]. The survey consisted of mostly multiple-choice questions and took approximately 20 min to complete.

### 2.4. Measures

#### 2.4.1. Delivery of Preventive Care

A total of 16 items were developed by the research team to collect data about current levels of preventive care provision. These items were adapted from items previously used by the research team to assess preventive care provision in clinical mental health [44,45], substance use [46], and other health services [47]. The items were also assessed for their suitability for use with staff by CMO representatives (co-authors: JR, MO, JF). Preventive care was defined as delivering a brief intervention, as per steps taken from the 5As and AAC models. The three steps “Ask”, “Advise”, and “Assist” were assessed to capture when support was delivered directly by the staff members, and the step “Connect” was assessed to capture when connections or referrals to external support services or health professionals were made. Each step (Ask, Advise, Assist, Connect) was assessed for each risk behaviour: smoking; low fruit and vegetable consumption; harmful alcohol consumption; and physical inactivity. For example, to assess the provision of care for physical activity the following four questions were asked: (1) Approximately what proportion of people accessing the service do you *ask* about their current physical activity? (2) Approximately what proportion of people accessing the service (who engage in health risk behaviours) do you provide brief *advice* to about increasing their physical activity? (3) Approximately what proportion of people accessing the service (who engage in health risk behaviours) do you *assist* with increasing their physical activity? (4) Approximately what proportion of people accessing the service (who engage in health risk behaviours) do you *connect* with a health professional or behaviour change support service for assistance with increasing their physical activity? The response options for all questions were (1) “Please provide an approximate percentage between 0 and 100”, (2) “unsure”, or (3) “prefer not to answer”.

#### 2.4.2. Barriers to Providing Preventive Care

A 69 item survey was developed by the research team using the TDF survey template [48]. The TDF’s 14 domains and item templates have been validated with care providers in various clinical and community-based settings [49,50]. In this study, the TDF item templates were adapted so that they could assess potential barriers and facilitators to the provision of preventive care by CMO staff to people with mental health conditions. Items developed for the domains “Knowledge” and “Skills” were subsequently merged into one domain based on previous research demonstrating that these two domains were highly correlated [48]. The 13 TDF domains and items per domain were as follows: Knowledge and Skills (16 items); Social and Professional Role Identity (3 items); Belief about Capabilities (8 items); Optimism (2 items); Belief about Consequences (3 items); Reinforcement (3 items); Intentions (1 item); Goals (1 item); Memory, Attention and Decision Processes (1 item); Environmental Context and Resources (23 items); Social Influences (3 items); Emotion (3 items); and Behavioural Regulation (2 items). A copy of all items in the survey can be seen in Appendix A. The response options were presented on a seven-point Likert scale from Strongly Disagree to Strongly Agree. Of the 69 total items, 22 were negatively worded to eliminate the effects of acquiescence bias, which can become problematic with agree-disagree response options [51]. The research team received feedback from CMO staff during the development of the survey to establish the face validity of the measure and ensure that the wording of items in the survey was acceptable.

#### 2.4.3. Staff Demographic Characteristics

Participants were asked questions regarding their: age; gender; Aboriginal and/or Torres Strait Islander identification; education level; professional qualifications; and whether they had a lived experience of a mental health issue. Participants were also asked about the location of the CMO site they worked in (postcode); their length of employment in their current role at the CMO; their employment status (full-time, part-time, casual); their current role within the CMO; and the programs and/or services within the CMO that they worked in.

### 2.5. Data Analysis

Prior to data analysis, data cleaning was performed to check for outliers. No missing data were imputed and the final sample was limited to participants that completed the TDF section of the survey. SPSS version 25.0 [52] was used to conduct data analysis. Descriptive statistics (frequencies and proportions) were used to report the demographic characteristics of participants in the study. The Accessibility and Remoteness Index of Australia (ARIA) [53] was used to categorise the location where staff worked (using the postcodes provided). The proportion of participants who answered that they provided each element of preventive care (ask, advise, assist, connect) to ≥80% of consumers for each health risk behaviour (smoking, fruit and vegetable consumption, alcohol consumption, physical activity) was calculated, to identify the proportion of current CMO staff providing “optimal” care. Participant responses to the questions in the TDF survey were scored from 1 to 7 where Strongly Disagree = 1 and Strongly Agree = 7 (with reverse scoring for items that were negatively worded). A mean score for each TDF domain was calculated by adding the score for each item and dividing it by the number of items in the domain. The closer the mean score was to 1, the more likely it was that the participants “disagreed” with the items in the domain and that this domain was a barrier to overall preventive care provision. The closer the mean score was to 7, the more likely it was that participants “agreed” with the items in the domain and that this domain was a facilitator of overall preventive care provision. Similar scoring methods have been applied to the TDF domains in previous preventive care research [44].

As the “connect” step was the element of preventive care least likely to be provided across all health behaviours, exploratory analyses were conducted to investigate whether particular TDF domains were associated with the provision of the “connect” element. Staff responses to the item “Approximately what proportion of people accessing the service (who engage in health risk behaviours) do you *connect* with a health professional or behaviour change support service for assistance” were dichotomised into “<80% of consumers” (not optimal) and “≥80% of consumers” (optimal).

Four separate logistic regression models were run for the four outcomes of interest: (1) Connect to support for smoking (not optimal/optimal); (2) Connect to support for fruit and vegetable consumption (not optimal/optimal); (3) Connect to support for alcohol consumption (not optimal/optimal); and (4) Connect to support for physical activity (not optimal/optimal). The overall mean scores for each of the 13 TDF domains were added as covariates to each of the four models. A backward stepwise approach was used for all four logistic regression models, with TDF domain variables removed until only those with *p* < 0.10 were retained. Odds Ratios (OR) and their 95% confidence intervals (CI) were calculated for each of the significant covariates (*p* < 0.05).

## 3. Results

Of the 634 potential participants who were identified and sent an email, 268 staff consented to participate (42%). Of the 268 staff who consented to participate, 34 did not meet the eligibility criteria. Of the 232 eligible staff, 190 (82%) answered items regarding barriers and facilitators to preventive care provision and were included in the study.

### 3.1. Staff Demographic Characteristics

The demographic characteristics of study participants are included in Table 1.

The majority of participants were female (70%) and identified as having a lived experience of a mental health issue (58%). A large proportion of participants had completed tertiary level education (Technical and Further Education (TAFE) 47% or University 47%) and had completed a professional qualification in mental health work (46%). Five percent of participants identified as Aboriginal and/or Torres Strait Islander. Most participants: had been employed with the CMO for <1 year (29%) or >2 years to 5 years (29%); were employed on a full-time basis (74%); in a person-centred support role (72%); and worked in a CMO service located in a major city (69%). The largest portion of participants reported working in accommodation support and outreach (42%) and life skill development (33%) programs. The mean age of participants was 42 years (SD = 12 years).

### 3.2. Delivery of Preventive Care

The proportion of staff providing preventive care to ≥80% of consumers to address their health risk behaviours, for each preventive care step, is presented in Table 2. A detailed report of preventive care delivery by CMO staff has been published elsewhere [54].

Asking and providing advice to ≥80% of consumers about their alcohol consumption was reported by a lower proportion of staff (44% and 38%, respectively) compared to asking and providing advice for other risk behaviours. Providing assistance to ≥80% of consumers for physical activity was most frequently reported by staff (45%). Less than one-third of staff reported providing connections to support for ≥80% of consumers at risk for any health behaviour (31% physical activity, 25% fruit and vegetable consumption, 18% alcohol consumption, 17% smoking).

### 3.3. Barriers to Providing Preventive Care

The raw scores for every item in the TDF survey are presented in Appendix A. The mean scores, medians, and range for each of the TDF domains are reported in Table 3. Behavioural regulation (M = 4.89, SD = 1.43), Optimism (M = 5.00, SD = 0.96), and Belief in Capabilities (M = 5.03, SD = 1.18) were the domains with mean scores closest to a minimum score of 1, indicating that staff was less likely agree with the items in these domains and these factors were more likely to act as potential barriers to overall preventive care provision.

### 3.4. Associations between the “Connect” Step of Preventive Care and TDF Domains

The results of the logistic regression models exploring the associations between the 13 TDF domains and the optimal provision of connections to consumers who required them (i.e., to ≥80% of consumers at risk) are presented in Table 4. There were no significant associations between the TDF domains and the optimal provision of connections for fruit and vegetable consumption or alcohol consumption.

As the mean scores for the “Knowledge and Skills” domain increased, the likelihood of providing a connection or referral to ≥80% of consumers who were current smokers also increased (OR = 2.55, CI = 1.01–6.47, *p* = 0.049). In contrast, as mean scores in the “Reinforcement” domain increased, the likelihood of providing a connection or referral to ≥80% of consumers who were smokers decreased (OR = 0.56, CI = 0.31–0.99, *p* = 0.046). The likelihood of providing a connection or referral for physical activity to ≥80% of consumers who required it increased as mean scores for the “Behavioural Regulation” domain increased (OR = 1.65, CI = 1.16–2.36, *p* = 0.006).

## 4. Discussion

This is the first study that has been conducted in a CMO setting where the TDF has been used to identify barriers to the provision of preventive care to consumers with mental health conditions. Alcohol consumption was least likely to be addressed for ≥80% of consumers for the Ask and Advise preventive care steps. Less than one-third of staff reported providing connections or referrals to support ≥80% of consumers at risk for any health behaviour. The TDF domains Behavioural Regulation, Optimism, and Belief about Capabilities had the lowest mean scores; indicating that these factors were more likely to be potential barriers to overall preventive care provision. Increasing scores in the Knowledge and Skills domain were associated with an increased likelihood of providing connections or referrals to ≥80% of consumers who were current smokers. In contrast, increasing scores in the Reinforcement domain were associated with a decreased likelihood of providing connections or referrals to ≥80% of consumers who smoked. Increasing scores in the Behavioural Regulation domain were associated with an increased likelihood of providing physical activity connections or referrals for ≥80% of consumers who required them.

### 4.1. Factors That May Be Acting as Barriers to Preventive Care Provision

Across the 13 domains assessed in the current study, the domain Knowledge and Skills had the highest overall mean score, indicating that most CMO staff had high agreement with items about their perceived knowledge and ability to ask, advise, assist and connect consumers with support to address chronic disease risk behaviours. High mean scores on the domains Beliefs about Consequences and Intentions also show that a large proportion of CMO staff perceive that providing preventive care: will benefit consumers’ physical and mental health; will not have a negative impact on their relationship with consumers; and that staff desire to improve their provision of preventive care. These findings are promising, as they suggest that CMO staff feel that they know how to deliver brief interventions to help modify health risk behaviours and perceive that doing so will have health benefits for the consumers they support.

However, lower levels of agreement for items in the domains of Behavioural Regulation, Optimism, and Belief about Capabilities were demonstrated by lower mean scores. CMO staff were less likely to agree that they felt confident providing preventive care; that they provided preventive care routinely; that it was not challenging to provide preventive care; and that they felt confident that consumers would follow through with connections or referrals they provided. This information is useful for CMOs, as it helps identify more specifically the issues that may need to be addressed to increase preventive care provision. For example, providing staff training that focuses on building confidence and skills in having difficult or challenging conversations (e.g., motivational interviewing skills) [55], or forums where staff are able to listen to consumers speak about their success in changing health behaviours may help to increase staff members perceived capabilities and optimism [56]. Additionally, CMOs could consider ways to support staff to deliver care more routinely using various strategies such as: monitoring care provision; providing assessment tools; or establishing pathways for incentives, rewards and feedback. Co-developing strategies to support preventive care provision with both CMO staff and consumers is recommended to ensure that any such strategies are acceptable and feasible, and likely to be implemented [57,58,59]. Further high-quality research is needed to evaluate whether such strategies (see Appendix A) are effective in improving CMO staffs’ provision of preventive care and subsequently whether this leads to improvements in consumer health risk behaviours.

### 4.2. Associations between the “Connect” Step and TDF Domains

In the regression analysis, a significant association was found between increasing scores in the Knowledge and Skills domain and providing connections to smoking cessation support for ≥80% of consumers who were at risk. These findings are similar to findings in other mental health settings, which have demonstrated that having practical skills in preventive care provision leads to increased care delivery. For example, a study conducted in a community mental health centre in the United States found that clinicians who had received formal training regarding how to counsel clients about cardiovascular disease risk factors (smoking, poor diet, and lack of exercise) had 2.7 times the odds of providing this care to more than 50% of their clients [60].

It is surprising that this association was only found for smoking, and not for other risk behaviours. Possible explanations could be a lack of local referral options to support smoking cessation, especially in non-metropolitan regions where health care specialists and behaviour change support services are less likely to be available [61]. It could be that CMO staff may not know about or know how to refer consumers to relevant telehealth or e-health services such as quitlines to support smoking cessation. Not knowing where to refer consumers for support has been identified in a previous study as a barrier to CMO staff providing care to consumers to quit smoking [36]. Alternatively, if staff are aware of support services, it might be that they perceive that existing services are not tailored to the needs and preferences of their consumers [26,62]. Further research to identify particular knowledge or skill gaps that CMO staff feel would assist them to increase the provision of connections to support smoking cessation is warranted.

An unexpected association was found between the provision of optimal connections for smoking and the TDF domain Reinforcement: as scores on the Reinforcement domain increased, the likelihood of providing connections for support with smoking decreased. The three items in the Reinforcement domain relate to feeling appreciated by consumers, being acknowledged by managers, and feeling a sense of satisfaction when delivering preventive care. These results suggest that positive opinions of the staff member by consumers and managers are not driving staff behaviour related to providing optimal connections for smoking, nor is personal satisfaction. Instead, these factors may be barriers to providing optimal care. Further research is needed with CMO staff to gain a better understanding of how internal and external reinforcement impacts preventive care provision.

An additional positive association was found between the provision of optimal connections for physical activity and increased scores on the TDF domain Behavioural Regulation. This finding suggests that staff who agreed that they provided care routinely and monitored the care they provided, were more likely to provide connections and referrals to ≥80% of consumers who were physically inactive. These results are supported by the implementation science literature, which reports that objectively measuring and providing feedback on the frequency of referrals or connections for chronic disease risk behaviours is an evidence-based way to increase them [63]. This type of strategy could be considered by CMOs as a potential way to increase the proportion of consumers that receive referrals to health specialists or connections to behaviour change support services.

### 4.3. Limitations

A limitation of this study was that data were only collected from staff representing a single CMO. To be able to generalise the findings of this research, a larger sample of staff representing more CMOs is required. However, the participating CMO had over 60 sites supporting service delivery across multiple states. Therefore, it is likely that reported preventive care provision across these different services can still provide an insight into the way in which CMO staff might address chronic disease risk behaviours with consumers. A further limitation is the consent rate achieved for the study, as only 40% of staff who were invited by email consented to participate. However, of those who did participate, 82% completed all questions related to barriers and facilitators to preventive care provision, and among the sample were staff who represented a diversity of demographic and employment-related characteristics. Finally, the TDF measure used to assess barriers and facilitators of preventive care provision may also be considered a study limitation. The survey was developed based on the TDF template [48]; however, the TDF itself is not a standardised measure, nor has it been validated in this context. The research team did however seek feedback from CMO staff during the development of the survey and prior to its administration, to establish face validity and ensure that the wording of survey items was acceptable.

## 5. Conclusions

This study identified several areas of potential barriers and facilitators to CMO staff providing preventive care to address health risk behaviours among consumers with mental health conditions. Promisingly, CMO staff reported knowing how to deliver preventive care and believing that such care would benefit consumers. However, barriers related to confidence in providing preventive care and in consumer uptake of referrals were observed among participating staff. Targeted training that takes these facilitators and barriers into account may be useful in improving the routine provision of preventive care by CMO staff. Further research among multiple organisations is needed to gain a more comprehensive understanding of preventive care practices and barriers to care provision in the Australian CMO sector. Conducting research with CMO staff at higher levels of the organisation (i.e., senior management staff) may also offer insights regarding the organisational features (i.e., guidelines, funding) that facilitate preventive care provision by direct support staff working on the ground.

## Figures and Tables

**Table 1 ijerph-19-04458-t001:** Demographic characteristics of CMO staff participants (N = 190).

Characteristic	Categories	N	%
Gender	Female	133	70
	Male	56	30
Identify as Aboriginal or Torres Strait Islander	No	178	95
	Yes	10	5.3
Highest Level of Education	University (CAE ^1^, Degree, or higher)	88	47
	TAFE ^2^ (Certificate, Diploma, Adv Diploma)	88	47
	Year 12 (Higher School Certificate) or less	13	6.9
Professional Qualification	Mental Health Work	87	46
	Peer Work	27	14
	Psychology	17	8.9
	Social Work/Welfare	8	4.2
	Nursing	7	3.7
	All other qualifications	30	16
	No qualification	14	17
Lived experience of a mental health issue	Yes	103	58
	No	74	42
Location of service where participant work	Major City	123	69
	Inner Regional	43	24
	Outer Regional/Remote Australia	12	6.7
Years of Employment at CMO	<1 year	49	29
	1 to 2 years	35	21
	>2 years to 5 years	49	29
	>5 years	36	21
Employment Status	Full Time	139	74
	Part Time/Casual	49	26
Current Role at CMO	Person centred support only	130	72
	Managerial only	25	14
	Both	25	14
Programs/Services that participant works in	Accommodation support and outreach	79	42
	Life skill development	62	33
	Holistic	50	26
	Self-help and peer support	38	20
	Leisure and recreation	34	18
	24-h support	26	14
	Promotion, information and advocacy	22	12
	Employment, education, training	20	11
	Family and Carer support program	9	4.7
	Clinical Service	2	1.1
	Helpline and counselling services	1	0.5
	Other program	14	7.4

^1^ College of Advanced Education. ^2^ Technical and Further Education. Ns for each item varied due to “prefer not to answer” responses.

**Table 2 ijerph-19-04458-t002:** Provision of care to ≥80% of consumers, by health risk behaviour and preventive care step.

Health Risk Behaviour	Preventive Care Step	Proportion of Staff Who Provided Care to ≥80% of Consumers	
		N	%
Smoking	Ask	86	58
	Advise	72	47
	Assist	38	25
	Connect	25	17
Fruit and Vegetable Consumption	Ask	78	51
	Advise	83	53
	Assist	56	36
	Connect	37	25
Alcohol Consumption	Ask	65	44
	Advise	58	38
	Assist	30	20
	Connect	27	18
Physical Activity	Ask	86	57
	Advise	90	58
	Assist	70	45
	Connect	46	31

**Table 3 ijerph-19-04458-t003:** CMO staff members mean and median scores for TDF domains.

TDF Domain	N	Mean	SD	Median	Minimum ^1^
Behavioural Regulation	181	4.89	1.43	5.50	1.00
Optimism	184	5.00	0.96	5.00	2.50
Belief about Capabilities ^2^	182	5.03	1.18	4.88	2.00
Environmental Context and Resources	181	5.05	0.82	5.09	1.50
Memory, Attention and Decision Processes	165	5.10	1.31	6.00	1.00
Social Influences	176	5.13	0.92	5.00	2.00
Emotion	175	5.14	1.08	5.33	2.00
Reinforcement	162	5.23	0.92	5.33	3.00
Goals	166	5.46	1.43	6.00	2.00
Social & Professional Role and Identity	179	5.64	0.99	5.67	1.00
Intentions	173	5.76	1.15	6.00	2.00
Belief about Consequences	183	5.88	0.87	6.00	2.50
Knowledge and Skills ^3^	186	6.11	0.86	6.22	2.75

^1^ Maximum score was 7 (“strongly agree”) for all domains. ^2^ The domain Belief about Capabilities (BC) had two questions per health behaviour. Means and SD by health behaviour are as follows: BC-Smoking (M = 4.90, SD = 1.30), BC-Fruit and vegetable consumption (M = 5.12, SD = 1.22), BC-Alcohol consumption (M = 4.96, SD = 1.24), BC-Physical activity (M = 5.15, SD = 1.24). The above items were included in calculating the overall domain score for Beliefs about Capabilities. ^3^ The domain Knowledge and Skills (KS) had four questions per health behaviour. Means and SD by health behaviour are as follows: KS-Smoking (M = 6.08, SD = 0.92), KS-Fruit and vegetable consumption (M = 6.17, SD = 0.86), KS-Alcohol consumption (M = 6.02, SD = 0.94), KS-Physical activity (M = 6.23, SD = 0.81). The above items were included in calculating the overall domain score for Knowledge and Skills.

**Table 4 ijerph-19-04458-t004:** Final logistic regression models exploring the associations between TDF domains (13 domains) and the optimal provision of connections for consumers with risk behaviours.

Provide Connections to >80% of Consumers with Risk Behaviours for:	Associated Domains	OR	95% CI	*p*-Value
Smoking	Knowledge and skills	2.55	(1.01, 6.47)	0.049
	Reinforcement	0.56	(0.31, 0.99)	0.046
Physical activity	Behavioural Regulation	1.65	(1.16, 2.36)	0.006

## Data Availability

The data are not publicly available due to protecting the confidentiality of study participants.

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
