# Peer review of "Barriers to the Provision of Preventive Care to People Living with Mental Health Conditions: Self-Report by Staff Working in an Australian Community Managed Organisation"

_ijerph, 2022, doi:10.3390/ijerph19084458_

Round 1

Reviewer 1 Report

Dear Editors,

Have a nice day!

This paper identifies the factors that might be acting as barriers to the provision of care services faced by community-managed organizations (CMOs) in order to assist people with mental health issues. This is a nicely drafted research paper. The introduction, methods, and analysis provide satisfactory information. However, I believe this paper is worth publishing if authors could add a graph showing barriers on one side and assisting measures on the other—a kind of framework out of their analysis based on theoretical domains framework (TDF). Rest is fine from my side.

Author Response

Response to Reviewer 1 Comments

Point 1: I believe this paper is worth publishing if authors could add a graph showing barriers on one side and assisting measures on the other—a kind of framework out of their analysis based on theoretical domains framework (TDF).

Response 1: Thank you for reviewing our manuscript and for your suggestion to include a graph with barriers and potential assisting measures suggested to address each barrier. We have included this graph as a supplementary file in the revised manuscript.

Reviewer 2 Report

The study is of course interesting, and the issue addressed by the research is relevant, because mental health has a great impact on quality of life of patients. In addition, as this paper points out, it is important to address the procedure that professionals carry out to improve the quality of life of patients. However, there are some major and minor issues to address

Abstract: Clear and concise.

Introduction:

At the beginning, the main measures to reduce the risk factors in this population are as follows (Ask, Assess, Advise, Assist, Arrange) [19-22] and the briefer AAC model (Ask, Advise and Connect) [23] (lines 50-51). However, how do CMOs operate, and are their guidelines based on these models or on others? It would be essential to specify this question.

It would be interesting to add studies that have been replicated on the performance of CMOs (or similar institutions) in other countries, as the identification of barriers seems to be a neglected issue.

There is no justification about the use of the Theoretical Domains Framework (TDF) in present research.

Materials and Methods:

Recruitment and data collection: Add any information about the lack of response or describe whether outliers have been removed from the data (participants contacted at the beginning, participants who finally access the studio).

Measures: What model or instrument did the team rely on to develop the 16-item questionnaire to assess the quality of current service delivery? It is specified that the research team relied on the 5As and AAC model, but it is not specified whether it is an ad-hoc questionnaire of their own invention or whether it is an adaptation of a previous questionnaire. If it is an adaptation, it would be advisable to mention the instrument on which this measure is based.

Data analysis: What software was used to perform the statistical analyses?

Results:

It is advisable that the information provided between lines 214-217 be included in recruitment and data collection.

Include the mean age of participants and standard deviation.

Table 3. Consider eliminating the table captions and including them in the information on the evaluation instruments.

References:

Check appropriateness of line 440.

Format line 495.

Line 500.

Line 502.

Line 508.

Line 518.

Line 531.

Author Response

Response to Reviewer 2 Comments

Point 1: At the beginning, the main measures to reduce the risk factors in this population are as follows (Ask, Assess, Advise, Assist, Arrange) [19-22] and the briefer AAC model (Ask, Advise and Connect) [23] (lines 50-51). However, how do CMOs operate, and are their guidelines based on these models or on others? It would be essential to specify this question.

Response 1: Thank you for your feedback. We have added the following information to clarify the ways in which CMOs operate:

Line 75: “There is some evidence that at least part of the 5A’s and AAC models form existing policies and guidelines for CMOs [34-36]. For example, one study of CMO leaders in New South Wales [34] found that smoking cessation support was provided to consumers through: assessment of smoking status, information about smoking cessation supports available (i.e., advise), and referral to external support with smoking cessation (i.e., connect). Additional support to assist consumers with smoking cessation was also reported, including support groups, Nicotine Replacement Therapy, and ‘SANE Smoke Free Kits’. However, across CMOs there is no uniform policy or guideline for addressing health risk behaviours and each individual organisation may design their own policies or guidelines.”  

Point 2: It would be interesting to add studies that have been replicated on the performance of CMOs (or similar institutions) in other countries, as the identification of barriers seems to be a neglected issue.

Response 2: Thank you for your comment. We have included the following information about international studies identifying barriers to CMOs addressing health risk behaviours:

Line 101: “Internationally, few studies have explored barriers to CMOs addressing health risk behaviours among people with mental health conditions [39,40]. One qualitative study of leaders of a U.S. psychiatric rehabilitation services ([40]comparable to CMOs in Australia) identified barriers to providing physical health support to include difficulties coordinating with external providers, using health IT, and finding staff who were competent and enthusiastic about providing physical health support to consumers. However, this study did not comprehensively assess possible barriers, and interviewed leaders of organisations rather than staff providing direct support to consumers. Only one study internationally has assessed CMO staff perceptions of facilitators or barriers to preventive care provision for a health risk behaviour [39]. This study focused on tobacco smoking and was conducted among 605 staff working across multiple CMOs in New Zealand. Factors identified as facilitating CMO staff to address smoking among consumers included: receiving training, having organisational policies, and having a staff culture that promoted positive lifestyle behaviours. Across previous research conducted within Australia and internationally, no study has comprehensively assessed staff reported barriers to the provision preventive care for multiple health risk behaviours for people living with mental health conditions.”

Point 3: There is no justification about the use of the Theoretical Domains Framework (TDF) in present research.

Response 3: Thank you for your feedback. We have added the following sentence to make the justification for using the TDF in our study clearer:

Line 124: “The TDF is useful for identifying areas of need for staff training, resource development, organisational culture, and guideline and policy development. This information can also inform the development of implementation interventions that may help to overcome reported barriers and increase preventive care provision in CMOs.”

Materials and Methods:

Point 4: Recruitment and data collection: Add any information about the lack of response or describe whether outliers have been removed from the data (participants contacted at the beginning, participants who finally access the studio).

Response 4: Thank you for your comment. We have added the following information to the ‘data analysis’ section of the paper:

Line 221: “Prior to data analysis, data cleaning was performed to check for outliers. No missing data was imputed and the final sample was limited to participants that completed the TDF section of the survey. SPSS version 25.0 [51] was used to conduct data analysis.”

Information regarding the study response rate (i.e., number of participants contacted, finishing survey etc.) has been retained in the ‘results’ section of the paper:

Line 258: “Of the 634 potential participants who were identified and sent an email, 268 staff consented to participate (42%). Of the 268 staff who consented to participate, 34 did not meet the eligibility criteria. Of the 232 eligible staff, 190 (82%) answered items regarding barriers and facilitators to preventive care provision and were included in the study.”

Point 5: Measures: What model or instrument did the team rely on to develop the 16-item questionnaire to assess the quality of current service delivery? It is specified that the research team relied on the 5As and AAC model, but it is not specified whether it is an ad-hoc questionnaire of their own invention or whether it is an adaptation of a previous questionnaire. If it is an adaptation, it would be advisable to mention the instrument on which this measure is based.

Response 5: Thank you for your feedback. We have added the following information to describe the design of the preventive care measures in more detail:

Line 166: “A total of 16 items were developed by the research team to collect data about current levels of preventive care provision. These items were adapted from items previously used by the research team to assess preventive care provision in clinical mental health [46,47], substance use [48], and other health services [49]. The items were also assessed for their suitability for use with staff by CMO representatives (co-authors: JR, MO, JF).”

Point 6: Data analysis: What software was used to perform the statistical analyses?

Response 6: The following information has been added to the ‘data analysis’ section of the paper:

Line 222: “SPSS version 25.0 [54] was used to conduct data analysis”

Results:

Point 7: It is advisable that the information provided between lines 214-217 be included in recruitment and data collection.

Response 7: Thank you for your suggestion. We have chosen to retain the information about participant response rates in the results section of the paper.

Point 8: Include the mean age of participants and standard deviation.

Response 8: Thank you for your suggestion. The mean age (and standard deviation) of participants is reported on Line 278. 

Point 9: Table 3. Consider eliminating the table captions and including them in the information on the evaluation instruments.

Response 9: Thank you for your suggestion. We have chosen to retain the information in the table captions.

Point 10: References: Check appropriateness of line 440. Format line 495, Line 500, Line 502, Line 508, Line 518, Line 531.

Response 10: Thank you for your attention to detail. We have updated the formatting of the above-specified references.

Reviewer 3 Report

I thank the authors and the editor for allowing me to read the manuscript “Barriers to the provision of preventive care to people living with mental health conditions: Self-report by staff working in an Australian Community Managed Organisation” I very much enjoyed reading this work. I believe that the findings of this study were very interesting and would certainly contribute to the understanding of these phenomena in the literature.

Here are my comments:

  1. This study presents the perception of the staff working in a cross-sectional study; it would be interesting to have evidence of how effectively the teams of the community managed organizations are received by the users, as there could be a gap between both perceptions.

  1. It would be interesting to be able to contrast whether there is indeed a change in the direction of greater adherence to care behaviors, for example, to have more objective measures of the care behaviors performed by users.

  1. An interesting variable that could imply a barrier or facilitator is self-description by the staff working on their care behaviors like do not smoke, it is easier to ask, advise, assist and connect.

  1. From the results, it would be interesting to identify more precisely how practices in community managed organizations can be improved.

Author Response

Response to Reviewer 3 Comments

Point 1: This study presents the perception of the staff working in a cross-sectional study; it would be interesting to have evidence of how effectively the teams of the community managed organizations are received by the users, as there could be a gap between both perceptions.

Response 1: Thank you for your suggestion. We have added a sentence to provide evidence of consumers perceptions regarding CMO services:

Line 67: “People with mental health conditions accessing CMO services (i.e. consumers) often rate their experiences with CMOs positively. One survey of consumers from 10 different CMOs in New South Wales, Australia, found that 89% of participants (N=1058) rated their experience with their CMO as ‘excellent’ or ‘very good’ [33].”

Point 2: It would be interesting to be able to contrast whether there is indeed a change in the direction of greater adherence to care behaviors, for example, to have more objective measures of the care behaviors performed by users.

Response 2: Thank you for your comment. We have added the following sentence in the discussion:

Line 380: “Further high-quality research is needed to evaluate whether such strategies (see supplementary figure 1) are effective in improving CMO staffs’ provision of preventive care and subsequently whether this leads to improvements in consumer health risk behaviours.”

Point 3: An interesting variable that could imply a barrier or facilitator is self-description by the staff working on their care behaviors like do not smoke, it is easier to ask, advise, assist and connect.

Response 3: Thank you for your comment. We agree that looking at CMO staffs’ own health risk behaviours and whether this impacts care provision to consumers would be an interesting factor to consider in future research.

Point 4: From the results, it would be interesting to identify more precisely how practices in community managed organizations can be improved.

Response 4: Thank you for your suggestion. We have added a supplementary figure that maps the three most prevalent barriers identified in the results to strategies that could be tested to improve practices in community managed organisations.

Round 2

Reviewer 1 Report

Dear Editors, 

Have a nice day. 

The authors have improved the quality of their work as per the given suggestion. It looks fine from my side.